# NEuroMOrphic Neural-Response Decoding System for Adaptive and Personalized Neuro-Prosthetics’ Control

**DOI:** 10.3390/biomimetics10080518

**Published:** 2025-08-07

**Authors:** Georgi Rusev, Svetlozar Yordanov, Simona Nedelcheva, Alexander Banderov, Hugo Lafaye de Micheaux, Fabien Sauter-Starace, Tetiana Aksenova, Petia Koprinkova-Hristova, Nikola Kasabov

**Affiliations:** 1Institute of Information and Communication Technologies, Bulgarian Academy of Sciences, 1113 Sofia, Bulgaria; georgi.rusev@iict.bas.bg (G.R.); svetlozar.yordanov@iict.bas.bg (S.Y.); simona.nedelcheva@iict.bas.bg (S.N.); aleksandar.banderov@iict.bas.bg (A.B.); 2Univ. Grenoble Alpes, CEA, Leti, Clinatec, F-38000 Grenoble, France; hugo.lafayedemicheaux@cea.fr (H.L.d.M.); fabien.sauter@cea.fr (F.S.-S.); tetiana.aksenova@cea.fr (T.A.)

**Keywords:** brain–machine interfaces, neural response decoder, spiking neural networks, neuromorphic systems, ECoG, personalized neuro-prosthetics

## Abstract

In our previous work, we developed a neuromorphic decoder of intended movements of tetraplegic patients using ECoG recordings from the brain motor cortex, called Motor Control Decoder (MCD). Even though the training data are labeled based on the desired movement, there is no guarantee that the patient is satisfied by the action of the effectors. Hence, the need for the classification of brain signals as satisfactory/unsatisfactory is obvious. Based on previous work, we upgrade our neuromorphic MCD with a Neural Response Decoder (NRD) that is intended to predict whether ECoG data are satisfactory or not in order to improve MCD accuracy. The main aim is to design an actor–critic structure able to adapt via reinforcement learning the MCD (actor) based on NRD (critic) predictions. For this aim, NRD was trained using not only an ECoG signal but also the MCD prediction or prescribed intended movement of the patient. The achieved accuracy of the trained NRD is satisfactory and contributes to improved MCD performance. However, further work has to be carried out to fully utilize the NRD for MCD performance optimization in an on-line manner. Possibility to include feedback from the patient would allow for further improvement of MCD-NRD accuracy.

## 1. Introduction

Recent advancements in brain–computer interface (BCI) technology significantly improved their reliability [1,2], providing revolutionary benefits to patients with neurological conditions, such as stroke, spinal cord injuries, and neuro-degenerative disorders [3,4]. In particular, the BCI applications to human prosthetics’ control demonstrated remarkable progress in the recent decade [5,6,7,8,9]. A pioneering work using ElectroCorticoGram (ECoG) measurements from the motor cortex in the brain to decode movement intentions of tetraplegic patients has been reported in [10,11,12]. The performance of the developed Motor Control Decoder (MCD) allows patients to use the neuro-prosthesis for several weeks without recalibration. However, an unsupervised recalibration/adaptation would provide great value for home use of such BCI-controlled devices.

For the aim of the auto-adaptation of MCD, feedback of how it performs on given task is beneficial. Different methods for assessing a person’s performance and for delivering feedback are used in different scenarios [6,8,9,13,14]. For the aim of MCD training, the patient was asked to imagine desired movement, and recorded ECoG signals were marked accordingly. However, since there is no guarantee that the patient is satisfied by the effector actions, the need for classification of brain signals as satisfactory/unsatisfactory was raised.

Very recent research [15] proposed a reinforcement learning (RL) framework for improving BCI performance. Since immersion of the subject in the task to be performed by a prosthetic device could not be completed during all periods of data collection, the brain signal decoder trained by data from less engaged trails could have lower accuracy. The authors proposed an RL-based decoder that performs continuous parameter updates in an on-line mode. In this framework, the neural activity from the medial prefrontal cortex was considered as a reward-related brain region that represents task engagement. The reported results demonstrate the potential of an RL framework for on-line BCI accuracy improvement.

Since the ECoG implants used in our study are positioned over the motor cortex, the question of whether we can extract useful information about motor decoder performance from this particular brain area arises. Ref. [16] reported that prediction errors are signaled in the ventromedial prefrontal and lateral orbitofrontal cortex, the anterior insula and dorsolateral prefrontal cortex. The authors suggest that these regions might therefore belong to brain systems that differentially contribute to the repetition of rewarded choices and the avoidance of punished choices. The investigation in [17] whether cortical responses measured by ECoG implants elicited from user error have the potential to serve as feedback to BCI decoders came to the conclusion that significant responses were noted in primary somatosensory, motor, premotor, and parietal areas of the brain. In search of principles for designing BMIs with learning abilities, Ref. [18] investigated neural information pathways among cortical areas in task learning and, particularly, the relationship between the medial prefrontal cortex and primary motor cortex, which are actively involved in motor control and task learning. The authors found that motor cortex spikes can be well predicted from spikes in the medial prefrontal cortex. Since the co-activation between these areas evolves during task learning, and becomes stronger as subjects become well trained, prediction of motor cortex activity from the activity in the prefrontal cortex could help in the design of adaptive BMI decoders for task learning.

In [13], a framework for the detection of neural correlates of task performance from ECoG implants positioned at the motor cortex area was investigated, and a database labeled with satisfactory and unsatisfactory brain signals was created. Each movement of the prosthetic device provoked by a patient’s brain signals and decoded by a well trained MCD was labeled as satisfactory or not satisfactory, depending on the resulting error of the controller (MCD). Only satisfactory brain signals were used to train the MCD. Then, another model for the prediction of the neural response quality, called Neural Response Decoder (NRD), was trained. A further aim of NRD was to allow for the auto-adaptation of the MCD in an on-line mode. However, since the initial MCD, which is supposed to be well trained, might not be perfect, such a labeling of ECoG data as satisfactory/unsatisfactory might not be always exact. Since our work is based on the database described in [13], we propose that the satisfaction labels could be considered as punish/reward signal to be exploited to train an actor–critic RL architecture.

Since the task of brain signal decoding cannot be solved by traditional linear models, intensive research towards the development of various nonlinear approaches is underway. A recent work [19] compared the power of nonlinear and linear approaches on a task for predicting finger movements. The authors addressed concerns about neural networks having the possibility of obtaining inconsistent solutions. The reported results, however, demonstrated that various artificial neural network architectures showed promising results and could be used in the future for BMI decoding. However, the powerful ANNs usually need a lot of computational and energy resources for training and adaptation, which prevents their application in real time. That is why, nowadays, numerous neuromorphic devices were developed [20,21,22,23,24,25,26,27,28] in order to implement ANNs in less energy-consuming and fast embedded devices.

In our previous work [29], a novel neuromorphic framework of a BMI system for prosthetics’ control via decoding ECoG brain signals was described. It includes a three-dimensional spike timing neural network (3D-SNN) for spatio-temporal brain signal feature extraction and an on-line trainable recurrent reservoir structure (Echo state network (ESN)) for Motor Control Decoding (MCD). The decoder auto-adapts to the incoming brain signals via Spike Timing Dependent Plasticity (STDP) of connections (synapses) in the 3D-SNN structure. Based on our previous work, we upgrade our neuromorphic MCD with an NRD that should be able to predict whether the decoder actions and/or ECoG data from the patient are correct (satisfactory) or not. The main aim is to design an actor–critic structure able to adapt via reinforcement learning MCD (actor) based on NRD (critic) predictions. It would allow for the continuous adaptation of the decoder to changes in brain signals in real time.

The contributions of this work are the following:The task for neural response decoding (NRD) for the aim of MCD improvement is presented as a reinforcement learning framework.The MCD (actor) and NRD (critic) are designed as neuromorphic structures combining a 3D-SNN structure for spatio-temporal feature extraction from ECoG implants and an on-line trainable ESN for the final decoding stage, which makes their implementation suitable for a neuromorphic chip, offering low power consumption, fast processing and small size.Several approaches to training both actor (MCD) and critic (NRD) in their interaction over time are investigated.Potential for on-line MCD improvement using NRD predictions is proven via simulations.

The rest of this manuscript continues with a section describing the experimental database, ECoG feature extraction, RL framework, neuromorphic NRD and MCD structures, and their software implementation. Next, training and testing algorithms are described, followed by the simulation results, their discussion and concluding remarks with directions for future work.

## 2. Materials and Methods

### 2.1. Experimental Data

The developed MCD-NRD structure was trained on a database (DB) called RUNNER. Figure 1 shows the experimental set-up.

During the data collection experiment, the patient was seated in front of a computer screen where a human avatar was represented from a third-person perspective. The avatar could either stand still or walk forward at a fixed speed. The RUNNER DB was collected from a patient with traumatic sensorimotor tetraplegia caused by a complete C4–C5 spinal cord injury and having two chronic wireless ECoG implants at the motor cortex area. ECoG signals were recorded thanks to the WIMAGINE implants [30], composed of 64 planar electrodes, at a sampling rate of 586 Hz. The amplitude of the ECoG signals was of the order of 200–300 µV, and they were pass-band filtered in the range from 0.5 Hz to 300 Hz. The patient was involved in the “BCI and Tetraplegia” clinical trial at CEA/Clinatec (NCT02550522), which focused on the recording and decoding of motor intentions with different effectors. Here, the patient controlled the avatar using leg motor imagery decoded by a trained MCD. In total, 9 sessions, of approximately 11 min of recording, were acquired over 99 days. For blind testing of the algorithms, the database was split in two parts: the first 7 sessions were used to train the decoder, while the last 2 sessions were left for testing the trained model’s accuracy.

The $State_{desired}$, i.e., the prescribed state of the screen avatar that should be predicted by MCD, has two possible values: idle or walk. The satisfaction of the decoded brain signals and achieved new state of the avatar (denoted briefly as SATIS) was marked according to the procedure described in [13,14], as follows: after a time lag, when the instruction on the screen was changed, if the avatar is in the $State_{desired}$ as a result of proper decoding of the patient’s ECoG signals by the MCD, this is considered satisfactory (SATIS=1); if it is not, the ECoG signal is marked as non-satisfactory (SATIS=0).

However, the way of labeling does not depend on the patient’s opinion. Since the avatar states were decoded by a pre-trained MCD module, the decoding error could result either from an incorrect patient’s brain signals due to fatigue and distraction, or from MCD incorrectness. Hence, such a labeling could serve to predict an error in desired movement decoding with an unknown origin.

Nevertheless, the experimental database is a good starting point to train initial NRD, which could be further refined using feedback from the patient.

### 2.2. Neuromorphic Framework for MCD and NRD

In our previous work [29], we described a neuromorphic structure called MCD. It was trained to predict a patient’s desired avatar actions from ECoG signals. Here, we upgrade it with an NRD structure whose aim is to predict correctness (satisfaction) of the actions decoded by the MCD. The basic idea comes from the reinforcement learning theory [31], which is considered a biologically plausible way that our brain learns from experience by interacting with the environment and receiving feedback about resulting outcomes. The feedback signal is called reinforcement and could be very simple, e.g., a binary label good=1 or bad=0.

Figure 2 presents the proposed reinforcement learning framework. In order to transfer the terminology from the experimental set-up to the terms in control theory and reinforcement learning, we refer to the patient as an object under control whose state ($object_{state}$) is assessed by the extracted features from the signals measured by the ECoG implants; the MCD will further be referred to as the actor (controller) that has to be optimized to generate an Action resulting in satisfaction (SATIS=1) rather than non-satisfaction (SATIS=0); the satisfaction itself is considered as a binary reinforcementsignal to be predicted by the critic element (in our scheme, the NRD). Table 1 summarizes the correspondence between the experimental and RL terms.

Thus, the MCD’s role is to generate an Action based on the perceived object’s state (ECoGfeatures), while the NRD should predict the outcome (SATIS) from the Action and perceived object’s state. In situations when the movement desired by the patient is not known, a perfectly trained NRD should be able support MCD in the generation of end effectors’ proper motion of the exoskeleton supporting the patient. However, since the labels in RUNNER DB described above were obtained without feedback from the patient, we cannot be sure whether incorrect avatar movement was due to fatigue or the distraction of the patient, or because the MCD was not perfectly trained. If feedback from a patient ($SATIS_{label\;from\;patient}$) is included in the experimental set-up, the RL framework will allow for the simultaneous training of the MCD and NRD.

The overall neuromorphic structure upgrades the MCD, as described in detail in [29] and as shown in Figure 3. It consists of the following basic modules:A filtering module that transforms the raw ECoG signals to input signals for 3D-SNN using Morlet wavelet transformation for multiple central frequencies and their combination into a feature matrix of the same size as the original one.A 3D recurrent SNN architecture called a 3D SNN cube, which is spatially structured and adaptable to an individual 3D brain template, is used for feature extraction from processed ECoG signals. It adapts continuously to the incoming input in unsupervised mode via the STDP rule.Two recurrent Echo state network (ESN) structures for decoding of the desired movement (MCD) and satisfaction (NRD) from extracted features (spiking frequencies of the selected neurons in the 3D-SNN module). It can be trained on-line in supervised mode via recursive least squares (RLS) or in an unsupervised regime via reinforcement learning (RL) rules.

In our approach, data from the 64 ECoG electrodes’ signals are treated in blocks (portions) of 59 points at each time step, corresponding to approximately 100 milliseconds of data recording, as in [13,14]. Since using wavelet transformations of electro-physiological signals is a commonly used approach for feature extraction, features extracted from each block in our work combine 15 Morlet wavelet transformations, having 15 different central frequencies (from 10 to 150 Hz with step size of 10 Hz). The choice of central frequencies is based on previous works [13,14]. The extracted ECoG features in Figure 3 are a matrix of the same size as the current ECoG data block, i.e., 59×64, as follows:(1)$$ECoG\;feature_{e}^{i}\left(k\right)=AUC\left(Morlet_{f=[10\;20\;\ldots \;150]}^{i}\left(k\right)\right),i=1÷59,e=1÷64$$

Here, *e* denotes the ECoG electrode number, *i* is the number of points in the block *k*, and *f* is the fundamental frequency of the Morlet wavelet. AUC is an abbreviation for the area under curve. The curve for each point *i* contains 15 values of 15 Morlet wavelets with central frequencies f=[1020…150] Hz.

This approach to ECoG feature extraction constitutes a novelty, making features extracted suitable for on-line adaptations of 3D SNN cube connections reflecting time changes in ECoG signals. Its detailed investigation is a subject of another work that demonstrated a significant increase in MCD accuracy. Here, the reported results on MCD accuracy are also significantly better in comparison to those reported in our previous work [29], a fact in favor of this novel feature extraction approach.

The ECoG features are fed into the 3D SNN cube as generating currents for a time period corresponding to the time of the block recording (approximately 100 ms) to each of the 64 neurons in the structure. The neurons’ firing rates for this stimulation period are equal to the extracted spatio-temporal features from ECoG signals that are further fed into the MCD and NRD modules. Both the actor and critic modules are fast trainable recurrent neural network structures called Echo state networks (ESNs) [32]. They consist of a randomly connected pool of neurons with a hyperbolic tangent nonlinear activation function and a linear readout with weights trainable via the least squares method.

The top part of Figure 4 shows the 3D SNN cube structure. It consists of 64 spiking neurons (simulated by the Leaky Integrate and Fire (LIF) model) positioned at the ECoG electrode positions in a 3D space. The synaptic connections between each pair of neurons were randomly generated with weights proportional to their distance in the 3D space. The positive connections are marked in red, while the negative ones are marked in blue. All synapses are plastic, i.e., their strength (weight) adapts continuously to the input signals via an associative learning rule called Spike Timing Dependent Plasticity (STDP). The auto-adaptation in the 3D SNN structure reflects the spatio-temporal dependence of ECoG signals, thus accounting for the positions of the ECoG electrodes and changes of the recorded signals with time. The bottom part of Figure 4 shows the initial and two snapshots (after training and after testing) of the connections weights in the 3D-SNN structure. The color of each dot corresponds to the magnitude of the connection weight between each pair of neurons, numbered from 1 to 64 on the x and y axes. We observe that the initial weights (leftmost plot) converge very fast to connectivity, corresponding to the model input signals and continuing with small adaptation changes during the training (middle plot) and testing (right plot) phases.

The MCD is trained to predict the desired movement (Action=idle/walk) of the prosthetic device (in current experiment the avatar on the screen), while the NRD is trained to predict whether the brain signal from the ECoG generates desired (SATIS=1) or incorrect (SATIS=0) movements. The output of the NRD is considered as a prediction of the reward/punish signal in the form of satisfaction/non-satisfaction. This signal should allow for an adjustment of the actor’s behavior (MCD predictions) so as to decrease its error.

### 2.3. Software Implementation

All modules are written in Python, version 3.8.9. The 3D-SNN is based on the NEST simulator library, version 3.3 [33], while the rest of the code exploits numpy, SciPy and other Python libraries for mathematical calculations. The software works in pseudo-on-line mode and makes readings from $DB_{edf}$ files, the next portion of 59 records from ECoG electrodes, desired decoder outputs ($Action_{desired\;from\;DB}$ and $SATIS_{label\;from\;DB}$) from the $DB_{beh}$ file, and generates $Action_{predicted}$ and $SATIS_{predicted}$ as outputs for the current time period. During training, the model parameters are adjusted in supervised mode; during testing, the model’s output is kept in csv files.

The parameters that are adjustable in the supervised mode are the output connection weights of the ESN modules. They are tuned incrementally with every new input/output training data pair via the recursive least squares (RLS) method. The 3D-SNN connection weights auto-adapt via the STDP rule continuously. The 3D SNN cube state represents the membrane potentials of all the neurons in the structure. All model parameters are kept in csv files.

## 3. Methodology

We investigated four training approaches and performed three testing experiments, which are described further below.

### 3.1. Training Approaches

The training algorithm is outlined in Algorithm 1. TA1, TA2, TA3 and TA4 denote the four training approaches.

In the case of non-satisfaction, since it is supposed that the MCD should generate the opposite state, we first perform the following two training approaches:Firsttrainingapproach is denoted henceforth as **TA1** (Figure 5): Use the desired state of the MCD from the DB denoted as $Action_{desired\;from\;DB}$ as a target for the MCD and input to the NRD no matter whether the training example is labeled as satisfactory or non-satisfactory.Secondtrainingapproach is denoted henceforth as **TA2** (Figure 6): Use the swapped desired state of the MCD denoted as $reverted\;Action_{desired\;from\;DB}$ (if idle, revert to walk, and vise versa) as a target for the MCD and input for the NRD if the training example is labeled as non-satisfactory in the DB ($SATIS_{label\;from\;DB}=0$).

Since the decoder will not be aware of exact target movement of the test subject in real time, it is important to be able to rely on predictions from the MCD to train and test the NRD. In order to start training of the NRD with a better-trained MCD, we also performed the following two training approaches:Step1: Use the Firsttrainingapproach (**TA1**) to train the initial models of both the MCD and NRD using only the first training session from the DB.Step2: For the rest of the training sessions, use the Thirdtrainingapproach denoted as **TA3** (Figure 7) or the Fourthtrainingapproach denoted as **TA4** (Figure 8).

The Third (**TA3**) and Fourth (**TA4**) training approaches exploit the idea from the previous once: to use the $Action_{desired\;from\;DB}$ or to invert it (reverted $Action_{desired\;from\;DB}$) based on the value of the $SATIS_{label\;from\;DB}$ for training of the MCD.

All training approaches rely on model target output data from the DB; therefore, they could be implemented in off-line mode. In real time, if the target output is available, the RLS algorithm will allow the user to adjust the model parameters in on-line mode too.
**Algorithm 1** Pseudo-code of training algorithm**Initialization**Initialize ESNMCD and ESNNRD module parameters
Compose 3D-SNN module using ECoG positions
Initialize the cube connection weights based on the neurons’ distances
      **while**
 newdata
**do**
  2:   $ECoG\;signals\;for\;time\;period\;t_{ECoG}\leftarrow Read\;from\;DB_{edf}\;file$   $Action_{desired\;from\;DB}\;and\;SATIS_{label\;from\;DB}\leftarrow Read\;from\;DB_{beh}\;file$  4:   filteredECoGsignals←filter(ECoGsignals)   3D−SNNinput←filteredECoGsignals  6:   **for** $t_{ECoG}$ **do**       Simulate3D−SNN  8:   **end for**   ECoGfeatures←spikingfrequenciesof3D−SNN10:   **if** MCD training **then**       $ESN_{MCD}\leftarrow ECoG\;features$12:       $Action_{predicted}\leftarrow ESN_{MCD}\;output$       **if** TA1 or TA3 **then**14:            $MCD_{error}=Action_{predicted}-Action_{desired\;from\;DB}$       **else if** TA2 or TA4 **then**16:            **if** $SATIS_{label\;from\;DB}=1$ **then**                $MCD_{error}=Action_{predicted}-Action_{desired\;from\;DB}$18:            **else if** 
$SATIS_{label\;from\;DB}=0$ **then**                $MCD_{error}=Action_{predicted}-reverted\;Action_{desired\;from\;DB}$20:            **end if**       **end if**22:       $new\;ESN_{MCD}\;parameters\leftarrow RLS\left(MCD_{error}\right)$
   **else if** NRD training **then**24:       **if** TA1 **then**            $ESN_{NRD}\leftarrow \left[ECoG\;features,Action_{desired\;from\;DB}\right]$26:       **else if** TA2 **then**            **if** $SATIS_{label\;from\;DB}=1$ **then**28:                $ESN_{NRD}\leftarrow \left[ECoG\;features,Action_{desired\;from\;DB}\right]$            **else if** $SATIS_{label\;from\;DB}=0$ **then**30:                $ESN_{NRD}\leftarrow \left[ECoG\;features,reverted\;Action_{desired\;from\;DB}\right]$            **end if**32:       **else if** TA3 **then**            $ESN_{NRD}\leftarrow \left[ECoG\;features,Action_{predicted}\right]$34:       **else if** TA4 **then**            **if** $SATIS_{label\;from\;DB}=1$ **then**36:                $ESN_{NRD}\leftarrow \left[ECoG\;features,\;Action_{predicted}\right]$            **else if** $SATIS_{label\;from\;DB}=0$ **then**38:                $ESN_{NRD}\leftarrow \left[ECoG\;features,\;reverted\;Action_{predicted}\right]$          **end if**40:       **end if**       $SATIS_{predicted}\leftarrow ESN_{NRD}\;output$42:       $NRD_{error}=SATIS_{predicted}-SATIS_{label\;from\;DB}$       $new\;ESN_{NRD}\;parameters\leftarrow RLS\left(NRD_{error}\right)$44:   **end if****end while**46:Keepmodelparameters


### 3.2. Testing Experiments

The testing algorithm is represented by Algorithm 2. TE1 and TE2 denote the First and Second test experiments, which are described further below.
**Algorithm 2** Pseudo-code of testing algorithm**Initialization**Set ESNMCD and ESNNRD module parameters to the trained ones
Compose 3D-SNN module using ECoG positionsSet 3D-SNN state to the achieved after trainingSet cube connection weights to the values achieved after training
       **while**
 newdata
**do**
  2:   $ECoG\;signals\;for\;time\;period\;t_{ECoG}\leftarrow Read\;from\;DB_{edf}\;file$   $Action_{desired\;from\;DB}\leftarrow Read\;from\;DB_{beh}\;file$  4:   filteredECoGsignals←filter(ECoGsignals)   3D−SNNinput←filteredECoGsignals  6:   **for** $t_{ECoG}$ **do**       Simulate3D−SNN  8:   **end for**   ECoGfeatures←spikingfrequenciesof3D−SNN10:   $ESN_{MCD}\leftarrow ECoG\;features$   $Action_{predicted}\leftarrow ESN_{MCD}\;output$12:   **if** TE1 **then**       $ESN_{NRD}\leftarrow \left[ECoG\;features,\;Action_{desired\;from\;DB}\right]$14:**   else if** TE2 **then**       $ESN_{NRD}\leftarrow \left[ECoG\;features,\;Action_{predicted}\right]$16:   **end if**   $SATIS_{predicted}\leftarrow ESN_{NRD}\;output$18:**end while**Keep3DSNNstateandcinnectionweights


In total, three testing experiments were carried out, as follows:Firstexperiment is denoted further as **TE1** (Figure 9): Feed the trained NRD with the desired action from the DB ($Action_{desired\;from\;DB}$) rather than from the trained MCD prediction. In this way, we skip the MCD imitating knowledge about instructions on the screen. However, in on-line mode, the NRD must know the target action, which is not always possible.Secondexperiment is denoted further as **TE2** (Figure 10): Feed the trained MCD prediction ($Action_{predicted}$) to the NRD, which is not always correct but will be available in a real situation. In this way, the decoder works fully in on-line mode.Thirdexperiment is denoted further as **TE3**: Testing of both models trained via the third (**TA3**) and fourth (**TA4**) training approaches was carried out as in the second experiment **TE2**, i.e., in on-line mode.

## 4. Results

For the training and testing of both the MCD and NRD modules, we use the fully labeled data from training sessions of RUNNER DB, i.e., sessions from 1 to 7 for training and the rest of the sessions (8 and 9) for testing.

Table 2 and Table 3 show the NRD testing accuracy from the Firstexperiment and the Secondexperiment for both models trained using the First and Second training approach, respectively. For the Firstexperiment, we observe that when the NRD was trained using information about the desired avatar movement from the instructions on the screen, it could be better trained to predict the SATIS label. In case where the desired actions were replaced by MCD predictions, as in the Secondtrainingapproach, the NRD’s accuracy dropped significantly in comparison to that which was expected, since the MCD was not perfectly trained.

The results of the Secondexperiment show that even if the model was trained using exact information about target movement, as in the Firsttrainingapproach, testing with trained MCD predictions yields decreased accuracy in comparison with the Firstexperiment. Again, the Secondtrainingapproach results in lower model accuracy.

Testing of both models trained via the Third (**TA3**) and Fourth (**TA4**) training approaches was carried out as in the Secondexperiment **TE2** (Figure 10). Table 4 shows the testing accuracy of the NRD from the third experiment. It is better than the accuracy achieved in the Secondexperiment and lower than the accuracy achieved in the Firstexperimet. The Fourthtrainingapproach yielded a model with better accuracy than expected.

Finally, we tested whether predictions from the trained NRD could be applied to improve the MCD’s accuracy, as shown by the dashed arrow from the NRD to the MCD in Figure 10. The results from the First and Second experiments are the same, since the MCD was trained in the same way in both cases. They are shown in Table 5. Table 6 shows the MCD’s accuracy from the Thirdexperiment.

We did not observe any significant differences in MCD accuracy in the First/Second and Third experiments. However, all experiments using training approaches TA2 (Second) and TA4 (Fourth), with reverting of $Action_{desired\;from\;DB}$ in the case of $SATIS_{label\;from\;DB}=0$, yielded better MCD accuracy in comparison with the First (TA1) and Third (TA3) training approaches (without reverting of $Action_{desired\;from\;DB}$). This proves the hypothesis that even if the NRD is not perfectly trained, its predictions can be applied to improve MCD’s accuracy when the decoder operates in real time.

## 5. Discussion

The results about NRD’s accuracy reported here demonstrate that if the NRD is given the target action to be performed by the MCD, it is able to distinguish correct from incorrect decoding of brain signals with much higher accuracy in comparison with the case when it is given the actual MCD’s output. Unfortunately, this approach could be applied only in off-line mode. However, when training both the NRD and MCD off-line first using knowledge about target actions and then continuing in on-line mode by feeding the NRD with output from the pre-trained MCD, the achieved NRD accuracy is significantly higher.

The testing results also demonstrated that even a non-perfectly trained NRD could improve the MCD predictions in a pseudo on-line mode if the MCD was trained with the reverting of $Action_{desired\;from\;DB}$ in the case of $SATIS_{label\;from\;DB}=0$, i.e., if we use only correct labels from the DB about the MCD’s desired output.

In order to be able to exploit the NRD predictions for the auto-adaptation of the MCD, the proposed RL framework needs to be applied in real time with the patient in the loop, as in Figure 2. This will allow us to communicate with the patient and observe changes in his/her brain signals, provoked by an adjustment in both the MCD and NRD. Thus, satisfaction labels would be much more correct since we would rely on a patient’s impression of how well the MCD predicts their desired action; at the same time, NRD’s accuracy could be improved based on patient feedback rather than off-line labeling in the DB. In real time, temporal difference (TD) learning [31] can be applied to make this model a true on-line algorithm. This will allow for a continuous adaptation of both the actor (MCD) and critic (NRD) networks. Further work in this direction using a true reinforcement learning algorithm will allow us to fully utilize the NRD for MCD’s performance optimization in an on-line manner.

## 6. Conclusions

The proposed neuromorphic NRD system inspired by reinforcement learning theory proves to be a good idea that could be explored further. Even though we did not have the opportunity to perform real-time experiments with patients, the collected experimental database allowed us to test several training and testing approaches. We demonstrated that if the training of the NRD starts with information about the correct target behavior of the MCD, this is a good starting point for further reinforcement training of both the MCD and NRD parts of the decoder.

Further experiments in real time with patients in the loop would allow us to obtain even better results. We believe that the patient must be asked about their satisfaction in order to achieve better auto-adaptation of their brain signals’ decoder during its exploitation in real-life situations. Reinforcement learning would allow us to continuously re-adjust the decoder in real time.

Overall, improving the performance of neuro-prosthetics for the benefit of a patient is an open problem. That is why a further study is still needed. While this paper covers a two-class prosthetic control problem, the general task comprises multiclass control of both legs and hands, as well as desired movement trajectory prediction. Since the team has collected databases from various experiments, our further work will be aimed towards designing a common on-line adaptable decoder of brain signals.

## Figures and Tables

**Figure 1 biomimetics-10-00518-f001:**
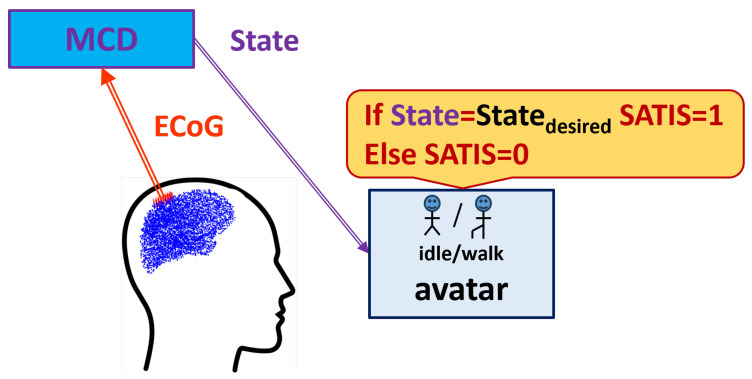
Experimental set-up for Runner DB acquisition.

**Figure 2 biomimetics-10-00518-f002:**
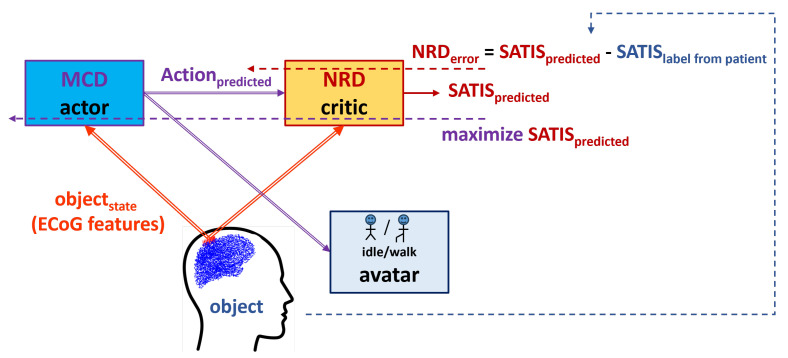
Actor–critic framework of the proposed reinforcement learning scheme.

**Figure 3 biomimetics-10-00518-f003:**
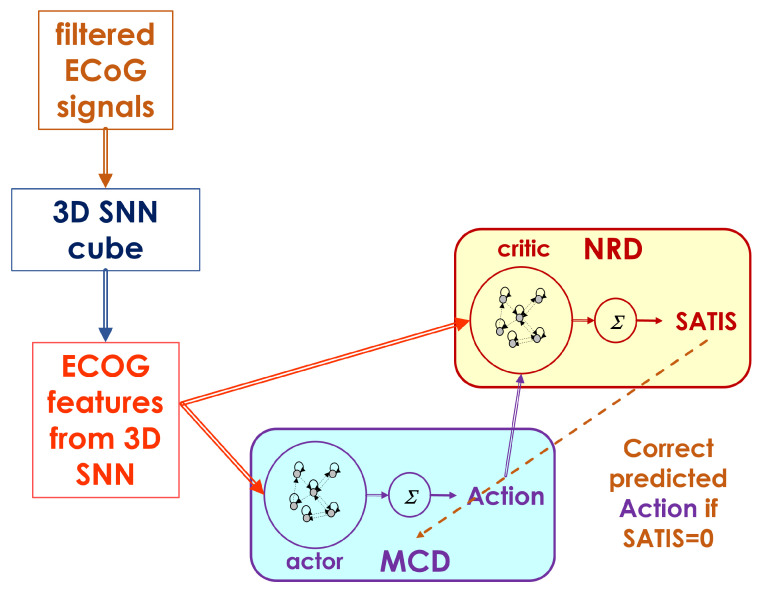
Decoder structure including both the MCD as actor and the NRD as critic network structures.

**Figure 4 biomimetics-10-00518-f004:**
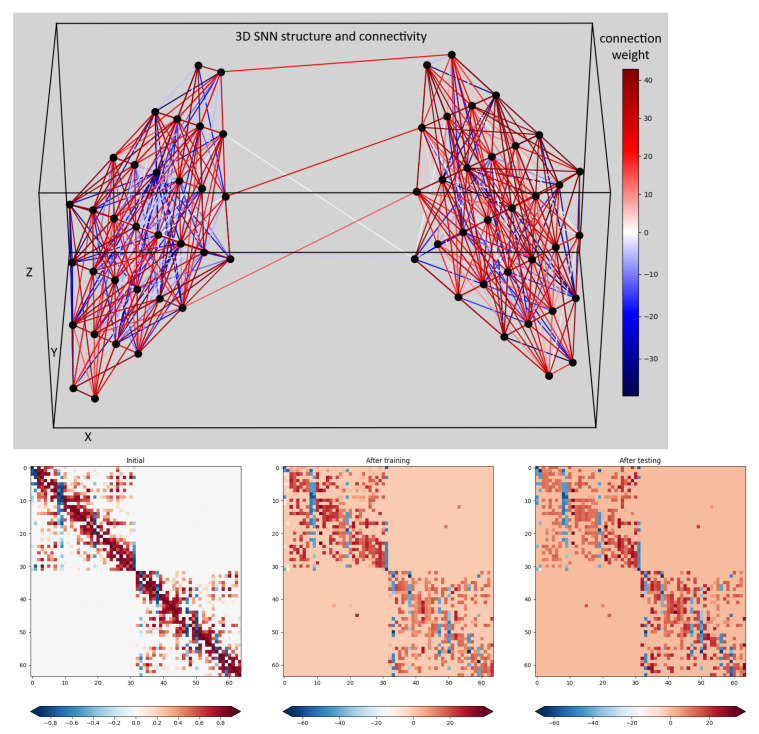
Connections in a 3D SNN.

**Figure 5 biomimetics-10-00518-f005:**
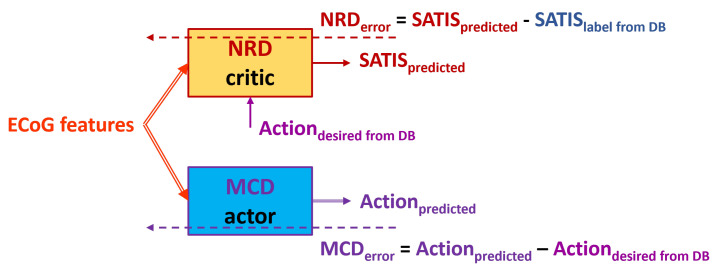
Block diagram of the Firsttrainingapproach **TA1**. Solid arrows represent the input and output data to/from both the NRD and MCD structures. Dashed arrows represent propagation of the training error for both the action and critic networks.

**Figure 6 biomimetics-10-00518-f006:**
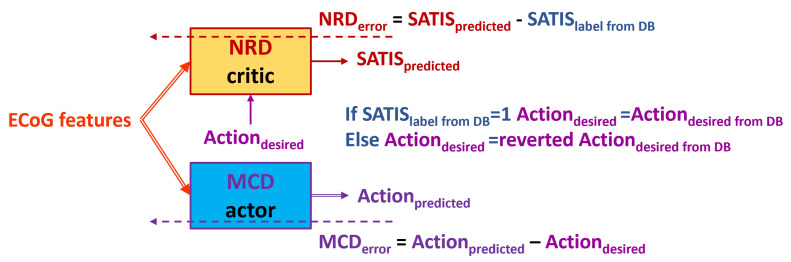
Block diagram of the Secondtrainingapproach **TA2**. Solid arrows represent the input and output data to/from both the NRD and MCD structures. Dashed arrows represent propagation of the training error for both the action and critic networks.

**Figure 7 biomimetics-10-00518-f007:**
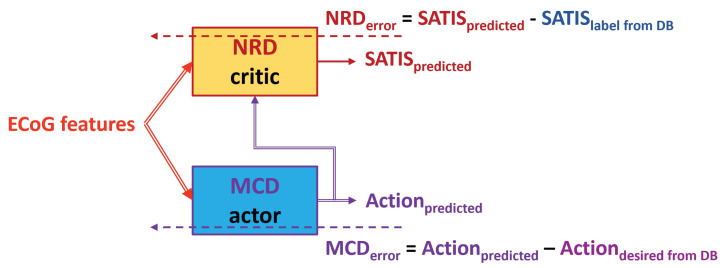
Block diagram of the Thirdtrainingapproach **TA3**. Dashed arrows represent training data and error, while the solid ones represent the input and output data to/from the MCD and NRD modules.

**Figure 8 biomimetics-10-00518-f008:**
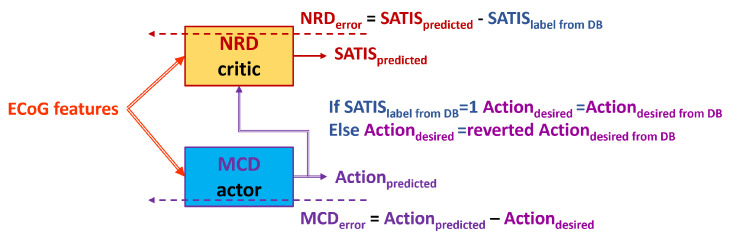
Block diagram of the Fourthtrainingapproach **TA4**. Dashed arrows represent training data and error, while the solid ones represent the input and output data to/from the MCD and NRD modules.

**Figure 9 biomimetics-10-00518-f009:**
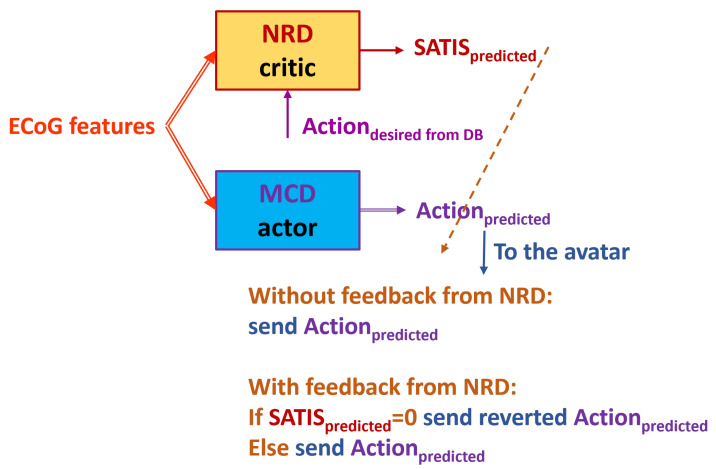
Testing in Firstexperiment **TE1**. The NRD uses known target movement of the avatar. The dashed arrow from the NRD’s output denotes possible corrective feedback from the NRD to the MCD’s output.

**Figure 10 biomimetics-10-00518-f010:**
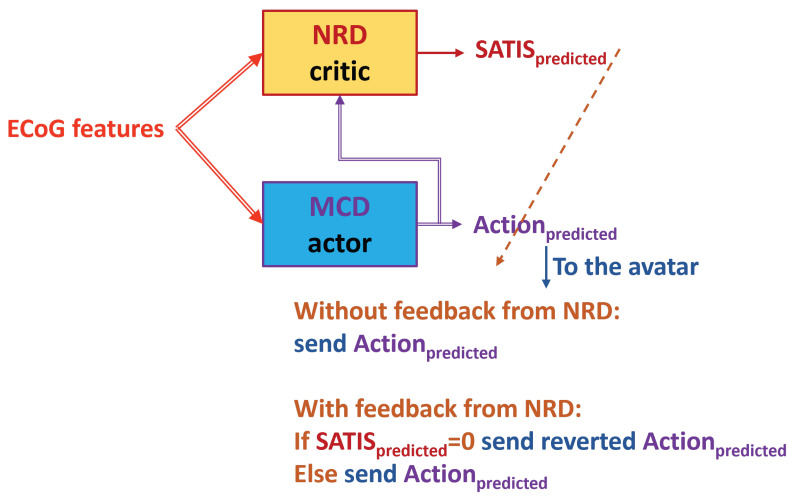
Testing in Secondexperiment **TE2**. The NRD uses predictions from the trained MCD. The dashed arrow from the NRD’s output denotes possible corrective feedback from the NRD to the MCD’s output.

**Table 1 biomimetics-10-00518-t001:** Terminology counterparts.

Experiment	Reinforcement Learning
Patient	Object
ECoG features	$object_{state}$
MCD	Actor
State	Action
NRD	Critic
Satisfaction	Reinforcement signal

**Table 2 biomimetics-10-00518-t002:** The testing accuracy of NRD from the Firstexperiment **TE1**. The best results are underlined.

Training Approach	Metrics	Session 8	Session 9
**TA1**	Balanced Accuracy	0.7491	0.7616
**TA2**	Balanced Accuracy	0.6474	0.5456
**TA1**	$F_{score}$ on SATIS=0	0.4145	0.4924
**TA2**	$F_{score}$ on SATIS=0	0.3643	0.1546
**TA1**	$F_{score}$ on SATIS=1	0.9219	0.9610
**TA2**	$F_{score}$ on SATIS=1	0.9491	0.9586

**Table 3 biomimetics-10-00518-t003:** The testing accuracy of the NRD from the Secondexperiment **TE2**. The best results are underlined.

Training Approach	Metrics	Session 8	Session 9
**TA1**	Balanced Accuracy	0.6279	0.5578
**TA2**	Balanced Accuracy	0.5291	0.4967
**TA1**	$F_{score}$ on SATIS=0	0.2796	0.1549
**TA2**	$F_{score}$ on SATIS=0	0.1386	0.0627
**TA1**	$F_{score}$ on SATIS=1	0.9177	0.9162
**TA2**	$F_{score}$ on SATIS=1	0.9028	0.9304

**Table 4 biomimetics-10-00518-t004:** The testing accuracy of the NRD from Thirdexperiment **TE3**. The best results are underlined.

Training Approach	Metrics	Session 8	Session 9
**TA3**	Balanced Accuracy	0.5637	0.5501
**TA4**	Balanced Accuracy	0.6787	0.6424
**TA3**	$F_{score}$ on SATIS=0	0.1800	0.1391
**TA4**	$F_{score}$ on SATIS=0	0.2761	0.2350
**TA3**	$F_{score}$ on SATIS=1	0.8657	0.8793
**TA4**	$F_{score}$ on SATIS=1	0.8532	0.9028

**Table 5 biomimetics-10-00518-t005:** The testing accuracy of the MCD from the First (**TE1**) and Second (**TE2**) experiments. The best results are underlined.

Training Approach	NRD Feedback	Metrics	Session 8	Session 9
**TA1**	YES	Balanced Accuracy	0.8069	0.7593
**TA1**	NO	Balanced Accuracy	0.8370	0.7699
**TA2**	YES	Balanced Accuracy	0.8723	0.8715
**TA2**	NO	Balanced Accuracy	0.8304	0.8251
**TA1**	YES	$F_{score}$ on walk	0.7861	0.7272
**TA1**	NO	$F_{score}$ on walk	0.8221	0.7393
**TA2**	YES	$F_{score}$ on walk	0.8389	0.8589
**TA2**	NO	$F_{score}$ on walk	0.8154	0.8086
**TA1**	YES	$F_{score}$ on idle	0.8285	0.7879
**TA1**	NO	$F_{score}$ on idle	0.8490	0.7973
**TA2**	YES	$F_{score}$ on idle	0.8767	0.8886
**TA2**	NO	$F_{score}$ on idle	0.8416	0.8415

**Table 6 biomimetics-10-00518-t006:** The testing accuracy of the MCD from the Thirdexperiment **TE3**. The best results are underlined.

Training Approach	NRD Feedback	Metrics	Session 8	Session 9
**TA3**	YES	Balanced Accuracy	0.7942	0.7691
**TA3**	NO	Balanced Accuracy	0.8370	0.7699
**TA4**	YES	Balanced Accuracy	0.8724	0.8800
**TA4**	NO	Balanced Accuracy	0.8366	0.8403
**TA3**	YES	$F_{score}$ on walk	0.7528	0.7363
**TA3**	NO	$F_{score}$ on walk	0.8221	0.7393
**TA4**	YES	$F_{score}$ on walk	0.8400	0.8693
**TA4**	NO	$F_{score}$ on walk	0.8209	0.8273
**TA3**	YES	$F_{score}$ on idle	0.8130	0.8047
**TA3**	NO	$F_{score}$ on idle	0.8490	0.7973
**TA4**	YES	$F_{score}$ on idle	0.8812	0.8945
**TA4**	NO	$F_{score}$ on idle	0.8508	0.8536

## Data Availability

No new data were created or analyzed in this study. This study presents only the MCD model’s structure and accuracy assessment obtained using simulations.

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
