# Peer review of "NEuroMOrphic Neural-Response Decoding System for Adaptive and Personalized Neuro-Prosthetics’ Control"

_biomimetics, 2025, doi:10.3390/biomimetics10080518_

Round 1

Reviewer 1 Report

Comments and Suggestions for Authors

Often theories (even the most logical ones!) are built on the basis of unmeasurable quantities. The same can be said about this work. In particular, all reasoning is built on the basis of some abstract ECoG Signal over a period of time. What exactly is meant: amplitude, if yes, then what specific frequency of waves, if a period of time, then what in ms or s?

What do 15 different fundamental frequencies mean?
Why are they FUNDAMENTAL?
Why a step of 10 Hz?
Why a range from 10 to 150 Hz?
WHAT ARE THE NEOPHYSIOLOGICAL BASIS for such a choice???

Often theories (even the most logical ones!) are based on immeasurable quantities. The same can be said about this work. In particular, all the reasoning is based on some abstract ECoG signal for a certain period of time. What exactly is meant: amplitude, if so, what specific frequency of waves, if a time interval, then in ms or s?
What do 15 different fundamental frequencies mean?
Why are they FUNDAMENTAL?
Why a step of 10 Hz?
Why a range from 10 to 150 Hz?
WHAT IS THE NEOPHYSIOLOGICAL BASIS for such a choice???
The explanation of the neurophysiological part of the study, based only on unpublished results and on own works, is unsatisfactory.

Author Response

Comment 1: Often theories (even the most logical ones!) are built on the basis of unmeasurable quantities. The same can be said about this work. In particular, all reasoning is built on the basis of some abstract ECoG Signal over a period of time. What exactly is meant: amplitude, if yes, then what specific frequency of waves, if a period of time, then what in ms or s?

Response 1: ECoG signals are recorded thanks to the WIMAGINE implants [29], composed of 64 planar electrodes, at a sampling rate of 586 Hz. The amplitude of the ECoG signals is of the order of 200-300 µV and pass-band filtered in the range from 0.5 Hz to 300 Hz. This is explained in previous works [13,14] as it is written on page 4, below Fig. 1 in the corrected manuscript.In our approach, data from the 64 ECoG electrodes' signals are treated by blocks (portions) of 59 points at each time step, which corresponds to approximately 100 ms of data. The text has been modified accordingly in green in the text on page 6.

Comment 2: What do 15 different fundamental frequencies mean? Why are they FUNDAMENTAL?

Response 2: The fundamental frequency of Morlet wavelet is its central frequency. Both terms are accepted in literature but for clarity we replaced the word “fundamental” by “central”.

Comment 3: Why a step of 10 Hz? Why a range from 10 to 150 Hz? WHAT ARE THE NEOPHYSIOLOGICAL BASIS for such a choice???

Response 3: We adopted the frequencies range from previous works [13,14] reporting results on the same data set as it is explained on page 6 because this choice is realistic from both neurophysiological way of how ECoG data was collected and from signal processing point of view. On page 6 of the revised manuscript details about ECoG features extraction are given pointing out that “Since using wavelet transformations of electro-physiological signals is a commonly used approach for features extraction, in our work features extracted from each block combine 15 Morlet wavelet transformations having 15 different central frequencies (from 10 to 150 Hz with step of 10 Hz). The choice of central frequencies is based on previous works [ 13 ,14].” The text further “This approach to ECoG features extraction constitutes a novelty making features extracted suitable for on-line adaptation of the 3D SNN cube connections reflecting the time changes in the ECoG signals. Its detailed investigation is a subject of another work that demonstrated significant increase of the MCD accuracy. Reported here results about MCD accuracy are also significantly better in comparison with those reported in our previous work [28], a fact in favor of this novel features extraction approach.” explains the novelty that however will be published in another work (still under review).

Comment 4: The explanation of the neurophysiological part of the study, based only on unpublished results and on own works, is unsatisfactory.

Response 4: We removed the unpublished work from the list of references. Besides, the introduction was extended by a review on neurophysiological foundations on which our work is based. The neurophysiological foundations were also explained in previously published works [13,14] where the experimental data base is taken from.

Reviewer 2 Report

Comments and Suggestions for Authors

The authors present an improved version of their previous approach called NEeuroMOorphic. After carefully reviewing, I found the following:

The authors intend to validated their approach, however, they insist that "further work has to be done to fully utilize the NRD for MCD performance optimization in on-line manner. Possibility to include feedback from the patient would allow for further improvement of MCD-NRD accuracy". Evidently, the real-world application could improve the performance of there neuromorphic approach. Based on this, how this improvement could improve the Neuro-Prosthetics Control?. 

Despide the authors presented some results, the main question is the following:

  • How these simulations could improve the performance of real Neuro-prosthetics approach?

State clearly the contribution of their approach by indicating its trade-offs.

Author Response

Comment 1: The authors intend to validated their approach, however, they insist that "further work has to be done to fully utilize the NRD for MCD performance optimization in on-line manner. Possibility to include feedback from the patient would allow for further improvement of MCD-NRD accuracy". Evidently, the real-world application could improve the performance of there neuromorphic approach. Based on this, how this improvement could improve the Neuro-Prosthetics Control?.Despide the authors presented some results, the main question is the following: How these simulations could improve the performance of real Neuro-prosthetics approach?

Response 1: In the discussion we’ve added explanations which algorithms can be applied in off-line or on-line mode. The following explanation how the simulations carried out could help to improve the Neuro-Prosthetics Control was added: “Thus, the satisfaction labels will be much more correct since we will rely on patients’ impression how well the MCD predicts his/her desired action and at the same time NRD accuracy could be improved based on patients’ feedback rather than from off-line labeling of DB. In real time the temporal difference (TD) learning [30] can be applied that is a true online algorithm. This will allow for continuous adaptation of both actor (MCD) and critic (NRD) networks.” We agree that real time experiment will be the best proof but since such an experiment has to involve the patient with disabilities it is difficult to arrange it. First, we have to prove that this could work by simulations. In concluding remarks we’ve added the following text explaining directions of our future work in this regard: “Overall, improving the performance of neuro-prosthetics for the benefit of a patient is an open problem and that is why a further study is always needed. While the paper covers a two class prosthetic control problem, the general task is a multiclass control for both legs and hands as well as desired movement trajectories prediction. Since the team has collected data bases from various experiments, our further work will be towards design of a common online adaptable decoder of brain signals.”

Comment 3: State clearly the contribution of their approach by indicating its trade-offs.

Response 3: In the introduction we’ve summarized our contributions as follows: The contributions of this work are:

  • The task for neural response decoding (NRD) for the aim of MCD improvement is presented as a reinforcement learning framework.
  • The MCD (actor) and NRD (critic) were designed as neuromorphic structures combining 3D-SNN structure for spatio-temporal features extraction from ECoG implants and an online trainable ESNs for final decoding stage, that makes their implementation suitable for a neuromorphic chip, offering low power consumption, fast processing and small size.
  • Several approaches to training of both actor (MCD) and critic (NRD) in their interaction over time are investigated.
  • Potential for online MCD improvement using NRD predictions was proven by simulations.

Round 2

Reviewer 1 Report

Comments and Suggestions for Authors

I am satisfied with the changes made
I recommend MS for publication

Comments on the Quality of English Language

I am satisfied with the changes made
I recommend MS for publication

Reviewer 2 Report

Comments and Suggestions for Authors

The paper can be accepted